# Impact of selective licensing schemes for private rental housing on mental health and social outcomes in Greater London, England: a natural experiment study

Jakob Petersen ,[1] Alexandros Alexiou,[2] David Brewerton,[3] Laura Cornelsen ,[1] Emilie Courtin,[1] Steven Cummins ,[1] Dalya Marks,[1] Maureen Seguin ,[1] Jill Stewart,[4] Kevin Thompson,[5] Matt Egan[1]

For numbered affiliations see end of article.

**Correspondence to**
Dr Jakob Petersen;
jakob.petersen@qmul.ac.uk

## ABSTRACT

**Objectives** To assess primary impact of selective Licensing (SL), an area-based intervention in the private rented housing market, on individual self-reported anxiety and neighbourhood mental health (MHI—Mental Healthcare Index) and secondary impacts on antisocial behaviour (ASB), population turnover and self-reported well-being.

**Design** Difference-in-difference (DiD) was used to evaluate effects of SL schemes initiated 2012–2018. 921 intervention areas (lower super output areas) were matched 3:1 using propensity scores derived from sociodemographic and housing variables (N=3.684 including controls). Average treatment effect on treated (ATT) was calculated for multiple time period DiD in area-level analyses. Canonical DiD was used for individual-level analysis by year of treatment initiation while adjusting for age, sex, native birth and occupational class.

**Setting** Intervention neighbourhoods and control areas in Greater London, UK, 2011–2019.

**Participants** We sampled 4474 respondents renting privately in intervention areas (N=17 347 including controls) in Annual Population Survey and obtained area-level MHI population data.

**Interventions** Private landlords in SL areas must obtain a licence from the local authority, allow inspection and maintain minimum housing standards.

**Results** ATT after 5 years was significantly lower for MHI (−7.5%, 95% CI −5.6% to −8.8%) than controls. Antidepressant treatment days per population reduced by −5.4% (95% CI −3.7% to −7.3), mental health benefit receipt by −9.6% (95% CI −14% to −5.5%) and proportion with depression by −12% (95% CI −7.7% to −16.3%). ASB reduced by −15% (95% CI −21% to −8.2%). Population turnover increased by 26.5% (95% CI 22.1% to 30.8%). Sensitivity analysis suggests overlap with effects of London 2012 Olympic regeneration. No clear patterns were observed for self-reported anxiety.

**Conclusions** We found associations between SL and reductions in area-based mental healthcare outcomes and ASB, while population turnover increased. A national evaluation of SL is feasible and necessary.

## STRENGTHS AND LIMITATIONS OF THIS STUDY

⇒ This is the first-ever evaluation of mental health and social outcomes of selective licensing schemes.
⇒ The multiple time period difference-in-difference design assesses impacts of the staggered area-based intervention over and above a host of other factors that influence mental health and well-being.
⇒ A limitation is that it is inherently not possible to eliminate selection bias due to non-random treatment allocation of selective licensing schemes.
⇒ As a limitation, the area-level findings of this study could not be complemented by individual-level data due to data sparsity in the survey sample.

## INTRODUCTION

Housing quality affects health.[1] Poor quality homes present numerous environmental risks to residents' health, including risks of injury, physical illnesses linked to cold, damp and indoor pollution, and risks to mental health and well-being.[2] The costs to the English healthcare system attributed to poor housing rivals those associated with hazards such as smoking and alcohol consumption[3 4]; costing an estimated £1.4bn in 2021.[4] The unequal distribution of poor-quality homes across the population correlates with other social inequalities in health.[5]

Housing improvement interventions can have a positive impact on residents' health, including mental health and well-being, particularly when targeted at those most in need.[2 6–9] Therefore, strategies for improving population health and health equity often include housing improvement.[1 10]

Housing quality improved between 2000 and 2019 in England across all sectors, but conditions are consistently worse in the private



rented sector (PRS) compared with owner-occupied and the social rented sector.[11] For instance, the proportion of homes failing to meet the criteria of the Decent Homes Standard in 2019 was 23% in PRS compared with 12% in the social rented sector and 16% for owner occupied homes. The PRS doubled between 2000 and 2019 in tandem with falling affordability of private homes and shrinking of the social housing sector.[11]

The need for action to improve PRS quality has been recognised by UK governmental bodies such as the National Audit Office[12] as well as the Chartered Institute for Environmental Health.[13] In 2006, local authorities gained discretionary powers to regulate privately rented homes through 'selective licensing' (SL) schemes under.[14 15] In SL schemes, landlords in areas targeted by local authorities must pay for a licence, allow inspection and carry out work necessary to maintain minimum housing standards. Fees are typically around £600 for a 5-year licence. SL schemes can only be implemented following a consultation with local stakeholders and only some local authorities have implemented SL to date.[15]

There are very few experimental and long-running studies of the links between housing and health due to lack of acceptability, ethics, treatment blinding and funding.[2 9] The evidence, therefore, mainly comes from observational, and often short term, studies of both individuals and neighbourhoods.[7 9] Although housing improvement interventions have on occasion been implemented as part of a randomised controlled study,[9] they are more typically implemented in ways that would require natural experimental impact evaluations.

A systematic review of the effect of housing improvement on health outcomes published in 2013 found the clearest evidence for interventions around thermal comfort, especially if targeted at people with the highest needs (poorer baseline health and/or socioeconomic status).[9] Being able to heat the home economically had positive impacts on health outcomes (general health, mental health, respiratory health, reduced absences from work and school) as well as facilitating better use of indoor space for the residents. In 2019, a systematic review of English-language studies from high-income countries found, in addition to heating, health benefits from improved ventilation, improved water supply and removal of indoor hazards.[2] Another recent review found evidence that mental health, well-being and other outcomes are at risk in the PRS, although the evidence base for interventions that might improve the sector was poor.[16]

Initially, government guidance on SL stated that schemes can be implemented to combat area-level problems such as antisocial behaviour (ASB).[17] The Housing Act 2004 stipulates that SL can only be implemented as a response to localised problems with low housing demand and persistent ASB.[16] ASB is defined in law as behaviours causing 'harassment, alarm or distress', which ranges from littering to complaints over rowdy neighbours.[18] New legislation enacted in 2015, however, gave local authorities wider powers to designate areas to SL based on poor housing conditions, high level of migration, deprivation and crime in addition to the previous conditions.[17] A survey of local authorities in 2019 found poor property conditions closely, followed by ASB as the most common reasons for introducing SL. Low demand (vacant housing), deprivation and crime were less commonly cited as reasons for introducing SLs and migration was rarely cited.[14]

A study commissioned by the Department for Levelling Up, Housing and Communities has described how local authorities vary their approach to regulating the PRS.[19] As the legislation allows some flexibility in how SL is implemented, there is scope for local authorities to tailor their SL to the local context and to addressing the reasons for introducing their scheme.An independent review found evidence that local schemes could vary their approach, along with a range of stakeholder views on potential mechanisms by which SL may affect ASB.[14]Although housing improvement interventions can lead to neighbourhood-level improvements,[7 9] the mechanisms by which SL may achieve such impacts (including ASB) are complex. SL schemes may include licence conditions that landlords take reasonable action to prevent and reduce ASB. Tenants may face eviction due to ASB and subsequently modify their behaviours, or be evicted. SL may also facilitate joint working across different agencies to tackle underlying issues associated with ASB, or assist policing, or provide training and support to encourage better standards in the sector.[14] We also hypothesise that improved property and positive feelings towards an area may link to reduced ASB. However, unintended impacts of SL, including potential harms, can also be hypothesised. For example, it is possible that costs for licence fees and required improvements are passed on to tenants, and leads to evictions. As a result, households experiencing hardships may be displaced to other localities or face homelessness. We will explore such mechanisms further in a subsequent paper based on qualitative data.

There have not been any systematic attempts to measure the potential impact of SL on mental health, well-being and ASB. This natural experiment study addresses this gap and functions as a feasibility study for a national evaluation of the impacts of SL. This paper primarily evaluates impacts on individual self-reported anxiety and neighbourhood mental healthcare in areas that have implemented SL compared with controls in Greater London. Secondarily, it evaluates self-reported well-being outcomes at the individual level, and ASB and population turnover at the area level.

## MATERIALS AND METHODS

A protocol paper describing the methodology in more detail has been published previously.[20] This paper concerns the quantitative outcomes of the protocol. The qualitative outcomes are currently being written up in a separate paper by the authors. Separate quantitative and

**Table 1** Selective licensing (SL) schemes in Greater London up until 2018 (year/local authority)

| Scheme | LSOA spatial units N | Population 2011 N | Treated private renters Annual mean (min; max) | Total N | Control private renters Annual mean (min; max) | Total N | Treated+ Total N |
|---|---|---|---|---|---|---|---|
| 2012 Newham | 155 | 291 351 | 110 (61;143) | 994 | 298 (175;393) | 2686 | 3680 |
| 2014 Barking-Dagenham | 110 | 185 911 | 66 (54;73) | 590 | 104 (83;132) | 937 | 1527 |
| 2015 Brent | 23 | 47 476 | | | | | |
| 2015 Waltham Forest | 144 | 258 249 | | | | | |
| 2015 Croydon | 220 | 363 378 | | | | | |
| 2015 Harrow | 7 | 11 653 | 156 (116;202) | 1406 | 549 (428;628) | 4938 | 6344 |
| 2016 Harrow | 6 | 11 394 | | | | | |
| 2016 Tower Hamlets | 22 | 38 354 | 22 (16;35) | 200 | 57 (25;82) | 511 | 711 |
| 2017 Ealing | 43 | 77 024 | | | | | |
| 2017 Redbridge | 16 | 28 789 | 31 (11;50) | 278 | 135 (74;181) | 1214 | 1492 |
| 2018 Harrow | 14 | 24 491 | | | | | |
| 2018 Brent | 42 | 75 793 | | | | | |
| 2018 Bexley | 13 | 23 499 | | | | | |
| 2018 Hackney | 15 | 26 366 | | | | | |
| 2018 Redbridge | 91 | 164 845 | 112 (74;141) | 1006 | 287 (225;337) | 2587 | 3593 |
| Total | 921 | 1 628 573 | – | 4474 | – | 12 873 | 17 347 |

Geographies were standardised to fully treated LSOA units. Population estimates are based on Census 2011. APS private renter responses in 2011–2019 tabulated by year of treatment initiation.
APS, Annual Population Survey; LSOA, lower layer super output aea.

qualitative papers allows for a more detailed descriptions of methods and findings from the two wings of the study.

## Patient and public involvement

We consulted two patient and public involvement representatives throughout the project.

## Interventions

We obtained details of the spatial and temporal extent of all current and historic SL schemes through freedom of information requests to all 33 local authorities in Greater London from when first enacted in 2006 to the end of 2019. We included all schemes initiated in or before 2018 in the analyses (table 1). To standardise the area-based data for analysis, conversion weights were calculated based on the number of 2011 Census enumeration postcodes[21] falling into small intercepts between the de facto geographical unit and the unit of analysis, lower layer super output areas 2011 (LSOA; approx. 1700 average population).[22] LSOA units that were only partially under treatment (conversion weights >0 and <1) were removed from both the treatment and control pool prior to analysis (N=17 LSOA excluded). Data from two boroughs that introduced street-level schemes (N=279 LSOA excluded), that is, Hammersmith and Fulham and Southwark, and a single electoral ward that was used as a pilot in Newham (N=9 LSOA excluded) were also excluded.

## Outcomes: area-level impacts

Small Area Mental Health Index (SAMHI) scores were obtained by year and small area (LSOA).[23] SAMHI combines data on mental healthcare from multiple sources into a single index, that is, National Health Service data on z-score standardised mental health-related admission (referred to as ADMISSION, hereinafter), antidepressant treatment days per population (PRESCRIPTION), primary care data on the percentage of the population diagnosed with depression (DIAGNOSIS) and Department for Work and Pensions data on the percentage of population in receipt of mental health-related benefits (BENEFITS). The SAMHI score is proportional to the overall burden on the healthcare system, that is, an increase signifies a worsening outcome. Each of the underlying SAMHI indicators (ADMISSION, PRESCRIPTION, DIAGNOSIS, BENEFITS) were, according to protocol, studied individually if a positive result was obtained with SAMHI itself.

High levels of ASB is one of the most common reasons for local authorities to implement SL,[14] so we assessed the incidence of police-recorded ASB by year and LSOA as a secondary outcome.[24] Data from a population turnover index were studied as a secondary outcome to test an association between SL exposure and moves.[25]

The population turnover index data are estimates based on a combination of electoral roll and consumer data (CDRC Residential Mobility Index 2020).[25] We include

the index as a proxy for changes in residential moves. The index is released as a cumulative and the annual proportion of households that will move in the coming year was derived for these analyses. The background for the index is the absence of officially released data other than the decennial censuses. The starting point for the index is the edited electoral roll (ie, the publicly available version without data on individuals who have opted out for privacy reasons and to avoid direct marketing) complemented with data on names and addresses of consumers collected by commercial data services companies.[25]

## Statistical methods: area-level impacts

A difference-in-difference (DiD) approach was deployed for the area-level impacts with three different strategies for controls: (1) All never-treated areas, (2) propensity score matched control (PSM) areas (the primary control strategy) and (3) not-yet-treated areas. The PSM controls were intended as a counterfactual based on measured baseline area characteristics, while the not-yet-treated controls, a counterfactual for unmeasured characteristics. Local authorities can justify the introduction of SL based on locally held data, for example, poor housing conditions. This is what we mean by the term unmeasured characteristics in these analyses. Never-treated controls were studied as a check of bias potentially introduced by the matching and trimming of the sample in PSM. The PSM used as far as possible preintervention sociodemographic, housing and neighbourhood characteristics from the 2011 Census, Indices of Multiple Deprivation and official dwelling age data (online supplemental table S1).[26–28] The matching was carried out with the Stata module KMATCH.[29] The parallel trend assumption was checked visually in the DiD plots.

Homeowners and social renters were by design studied in parallel with private renters for falsifiability checks. SL should only directly affect private renters and any effects detected for private renters could, therefore, also be challenged by studying not directly affected groups in the same intervention areas. Given the staggered nature of the intervention, a DiD method for comparing multiple time periods were used.[30] The number of intervention LSOA units was 921 and the total number of LSOA in the DiD-PSM analysis was 3684 (including three controls per one intervention area) (table 1). The average treatment effect on the treated (ATT) estimated by the DiD was given as ATT% for Ln-transformed indicators (BENEFITS, ADMISSION, ASB), $ATT\% = -100*(1-\exp(ATT))$. ATT% was for comparison also calculated for untransformed variables relative to the baseline value.

## Outcomes: individual-level impacts

Data on adult respondents in Annual Population Survey (APS) in England, 2011–2019, were obtained from Office for National Statistics (ONS).[31] Among these, we identified 4474 private renters exposed to the intervention (total number of renters including controls, N=17347) (table 1). The four subjective health and well-being

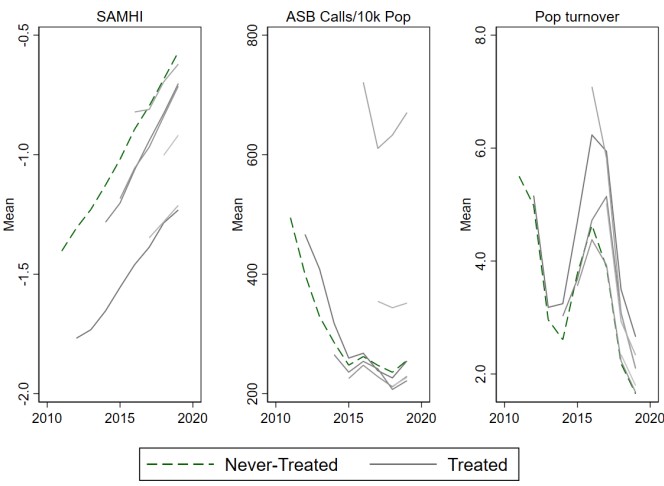

**Figure 1** Trend in area-level outcomes for never-treated versus treated areas in Greater London, 2011–2019. Treated areas shown from year of initiation onwards. ASB, antisocial behaviour; Pop, Population; SAMHI, Small Area Mental Health Index.

questions in APS (aka. ONS4) with scores from 0 to 10 were assessed. The anxiety question was the primary outcome and the other questions on subjective well-being (happiness, life satisfaction, whether the things you do in life are worthwhile), secondary outcomes. Data on how long the respondent had lived at the address (asked in categories and recoded to mid-category values for these analyses) were studied at the same time as a proxy of residential stability.

## Statistical methods: individual-level impacts

A canonical DiD approach was deployed for the individual-level impacts by year of treatment initiation in 2012, 2014 and 2015, respectively.[32] Schemes introduced the same year were pooled for statistical efficiency. Three different controls were used: (1) never-treated, (2) PSM controls and (3) PSM adjusted for age, sex, native birth and occupational class.[33]

## RESULTS

The size of the different SL schemes in terms of fully treated LSOA units, population and number of private renters captured in the APS data can found in table 1.

The overall trend in the composite mental healthcare indicator, SAMHI, was a gradual increase in burden during 2011–2019, while ASB calls declined sharply in 2011–2015 and then more slowly for most control and treatment groups (figure 1). Population turnover fluctuated during the study period. The trends for the underlying SAMHI indicators are shown in online supplemental figure S1.

The trends for the APS outcomes showed a slight improvement with a decline in how anxious the respondent felt the day before the interview and a slight increase for the other subjective well-being indicators (happy, satisfied, worthwhile) and years at address. The trends for

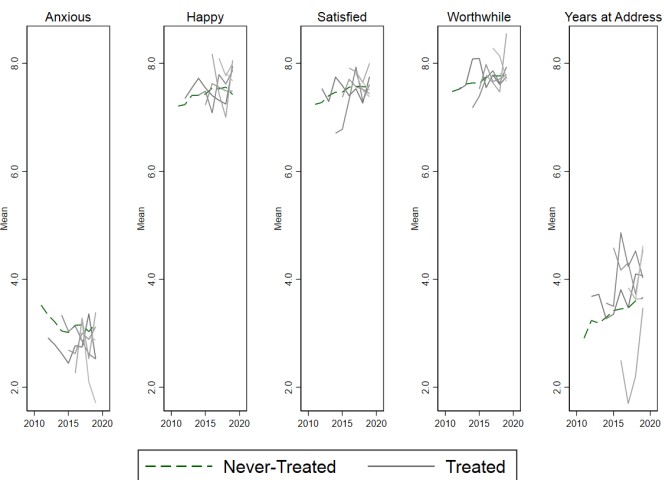

**Figure 2** Trend in individual-level outcomes for never-treated versus treated areas. Treated areas shown from year of initiation onwards.

the different SL schemes by year of treatment initiation were similar yet noisier presumably due to small number issues in the APS sample (figure 2).

The ATT with PSM controls after 5 years of intervention was significantly different from baseline for all area-based outcomes, SAMHI, ASB calls and population turnover (table 2, figure 3). Further analysis of the underlying SAMHI indicators showed similar positive results for antidepressant prescribing, depression diagnosis and mental health-related benefits, while no clear patterns were seen with mental health-related hospital admission (online supplemental figure 2). The average number of antidepressant treatment days per population in treatment areas at baseline was 13.1. This number reduced by −0.71 days (95% CI −0.95 to −0.48) after 5 years of intervention (table 2), that is, a −5.4% (95% CI −3.7% to −7.3%) reduction from the baseline in relative terms. Mental health-related benefits were received by 2.4% of the population at baseline and reduced by −9.6% (95% CI −14% to −5.5%), that is, −0.23 (95% CI −0.13 to −0.34) percentage point change in absolute terms. The proportion of the population diagnosed with depression was 3.5% at baseline and reduced by −0.42 percentage points (95% CI −0.57 to −0.27), that is, −12% (95% CI −7.7% to −16.3%) reduction of baseline in relative terms. ASB calls per 10 000 population were 537 at baseline and reduced (ie, improved) by −15% (95% CI −21% to −8.2%). Population turnover, as in the proportion of household that will move in the coming year, was 5.2% at baseline and increased by 1.38 percentage points, that is, 26.5% (95% CI 22.1% to 30.8%) in relative terms.

A sensitivity check of excluding the sole scheme initiated in 2012 was carried out. Apart from being the earliest London scheme, it also concerned the borough that was centre for the 2012 London Olympics (we here term it the 'Olympic' scheme). The results showed no 5-year results with SAMHI, similar reduction in ASB calls and a more modest increase in population turnover (online supplemental figure 3).

There were no clear patterns from the individual-level analyses of APS data (figure 4, online supplemental figures 4–7).

## DISCUSSION

The study found improvements in area-based mental health outcomes and ASB calls, while population turnover increased. Conversely, the results for self-reported anxiety and other individual-level indicators were inconclusive due to the small sample size of the APS data.

The results indicate potential benefits of SL schemes beyond their 5-year cycle, especially for reduction of ABS. We cannot exclude that at least part of the change could be due to gentrification and we saw an increase in population turnover to suggest this. Future quantitative studies of area-based impacts should therefore assess whether gentrification effects can be ruled out. Several mechanisms could potentially be at play. SL may encourage better practice among landlords and lead to improvements that may be sustained. Alternatively, SL may result in more landlords selling their properties rather than facing the increased cost burden, unregulated rentals, passing costs onto tenants through rent increases and evicting tenants with ASB behaviours with the opportunity to increase rents in high-demand areas. These hypothesised explanations are not mutually exclusive. Furthermore, the mechanisms at play may vary by scheme given differences in local context and given that the legislation allows for some flexibility in local delivery.

An interesting feature of these findings is that some of the changes in outcomes occurred before the completion of the 5-year licensing periods. This suggests the possibility that SL schemes may have impacts prior to full implementation. This could be important, as levels of enforcement may vary across London schemes: while there has not been a robust evaluation of this issue, the website www.londonpropertylicensing.co.uk provides some information on varying levels of enforcement based on periodic data requests from London local authorities.[15]

These first findings may be confounded by the fact that the earliest scheme overlapped with urban regeneration projects in connection with the 2012 London Olympics. A sensitivity check excluding the 'Olympic' scheme (Newham) did not show any reduction in the main area-based mental healthcare indicator, SAMHI. There was, however, a similar reduction in ASB and a more modest increase in population turnover after 5 years (both statistically significant). Studies of the impacts of the Olympic event itself and its legacy have notably been mixed. A telephone survey of residents in London, Berlin and Paris in 2011–2013 found a short-lived increase in subjective well-being for Londoners during the event.[34] A longitudinal cohort study of adolescents and their families living close to the Olympic site compared with those living further away found no changes in self-reported health behaviours or health outcomes (including subjective well-being) from before to 18 months after the event.[35] Co-occurring

**Table 2** Average treatment effect on treated (ATT) for area and individual impacts after 3, 5 and 7 years with PSM controls

| Indicator | Unit | Baseline mean (2011) | | ATT | | | ATT% |
| | | Never-treated | Treated | 3 years | 5 years | 7 years | 5 years |
|---|---|---|---|---|---|---|---|
| Area impacts—interventions initiated 2012–2018 | | | | | | | |
| SAMHI | Index score | −1.4 | −1.6 | **−0.03 (−0.05 to −0.02)** | **−0.12 (−0.14 to −0.09)** | **−0.27 (−0.29 to −0.24)** | **−7.5% (−5.6 to −8.8)** |
| Prescription | Antidepressant treatment days per pop | 15.5 | 13.1 | **−0.19 (−0.33 to −0.04)** | **−0.71 (−0.95 to −0.48)** | **−1.81 (−2.13 to −1.49)** | **−5.4% (−3.7 to −7.3)** |
| Benefits | %pop | 2.5 | 2.4 | **−8.5% (−11 to −5.8)** | **−9.6% (−14 to −5.5)** | −4.3% (−10 to 2.3) | **−9.6% (−14 to −5.5)** |
| Diagnosis | %pop | 4.3 | 3.5 | **−0.17 (−0.26 to −0.08)** | **−0.42 (−0.57 to −0.27)** | **−1.5 (−1.62 to −1.37)** | **−12% (−7.7 to −16.3)** |
| Admission | z-score | −0.71 | −0.54 | 24% (−11 to 72) | **−44% (−66 to −9.9)** | −23% (−58 to 40) | **−44% (−66 to −9.9)** |
| ASB | Calls per 10k pop | 495 | 537 | −3.8% (−7.8 to .41) | **−15% (−21 to −8.2)** | **−12% (−22 to −1.2)** | **−15% (−21 to −8.2)** |
| Pop turnover | %households moving+1 year | 5.5 | 5.2 | **0.39 (0.29 to 0.5)** | **1.38 (1.15 to 1.6)** | **0.86 (0.57 to 1.14)** | **26.5% (22.1 to 30.8)** |
| Individual impacts—interventions initiated 2012 | | | | | | | |
| Anxious | 0–10 scale | 3.5 | 4.2 | 0.09 (−0.72 to 0.9) | 0.35 (−0.5 to 1.21) | 0.59 (−0.28 to 1.45) | 8.3% (−11.9 to 5) |
| Happy | 0–10 scale | 7.2 | 7.1 | −0.14 (−0.78 to 0.5) | **−0.68 (−1.36 to −0.005)** | 0.03 (−0.62 to 0.68) | **−9.6% (−19.2 to −0.1)** |
| Satisfied | 0–10 scale | 7.2 | 7.2 | −0.31 (−0.83 to 0.2) | **−0.65 (−1.2 to −0.01)** | −0.24 (−0.78 to 0.29) | **−9% (−16.7 to −0.1)** |
| Worthwhile | 0–10 scale | 7.5 | 7.4 | 0.2 (−0.3 to 0.7) | −0.4 (−0.99 to 0.18) | −0.48 (−1.02 to 0.06) | −5.4% (−13.4 to 2.4) |
| Years at address | Years | 2.9 | 3.6 | 0.2 (−0.79 to 1.12) | 0.16 (−0.95 to 1.27) | 1.1 (−0.03 to 2.23) | 4.4% (−26.4 to 35.3) |
| Individual impacts—intervention initiated 2014 | | | | | | | |
| Anxious | 0–10 scale | 3.5 | 3.2 | −1.05 (−2.35 to 0.26) | −0.001 (−1.29 to 1.28) | N/A | 0% (−40.3 to 40) |
| Happy | 0–10 scale | 7.2 | 7.2 | 0.02 (−0.99 to 1.02) | −0.41 (−1.44 to 0.63) | N/A | −5.7% (−20 to 8.8) |
| Satisfied | 0–10 scale | 7.2 | 7.1 | **1.13 (0.32 to 1.93)** | 0.4 (−0.44 to 1.25) | N/A | 5.6% (−6.2 to 17.6) |
| Worthwhile | 0–10 scale | 7.5 | 7.6 | −0.05 (−0.82 to 0.71) | 0.31 (−0.5 to 1.11) | N/A | 4.1% (−6.6 to 14.6) |
| Years at address | Years | 2.9 | 2.7 | −1.48 (−2.99 to 0.03) | −0.55 (−2.01 to 1) | N/A | −20.4% (−74.4 to 37) |
| Individual impacts – Intervention initiated 2015 | | | | | | | |
| Anxious | 0–10 scale | 3.5 | 3.3 | 0.71 (−0.03 to 1.44) | N/A | N/A | N/A |
| Happy | 0–10 scale | 7.2 | 7.5 | −0.19 (−0.73 to 0.35) | N/A | N/A | N/A |
| Satisfied | 0–10 scale | 7.2 | 7 | −0.15 (−0.62 to 0.31) | N/A | N/A | N/A |
| Worthwhile | 0–10 scale | 7.5 | 7 | 0.01 (−0.48 to 0.5) | N/A | N/A | N/A |
| Years at address | Years | 2.9 | 2.7 | −0.65 (−1.64 to 0.33) | N/A | N/A | N/A |

ATT given as ATT% for Ln-transformed indicators (benefits, admission, ASB). For individual impacts, ATT adjusted for time-varying sociodemographic covariates and relate to the interventions initiated in 2012, 2014 and 2015. ATT values significant at 5% alpha level shown in bold face.
ASB, antisocial behaviour calls; N/A, not applicable; pop, population; PSM, propensity score matching; SAMHI, Small Area Mental Health Index.

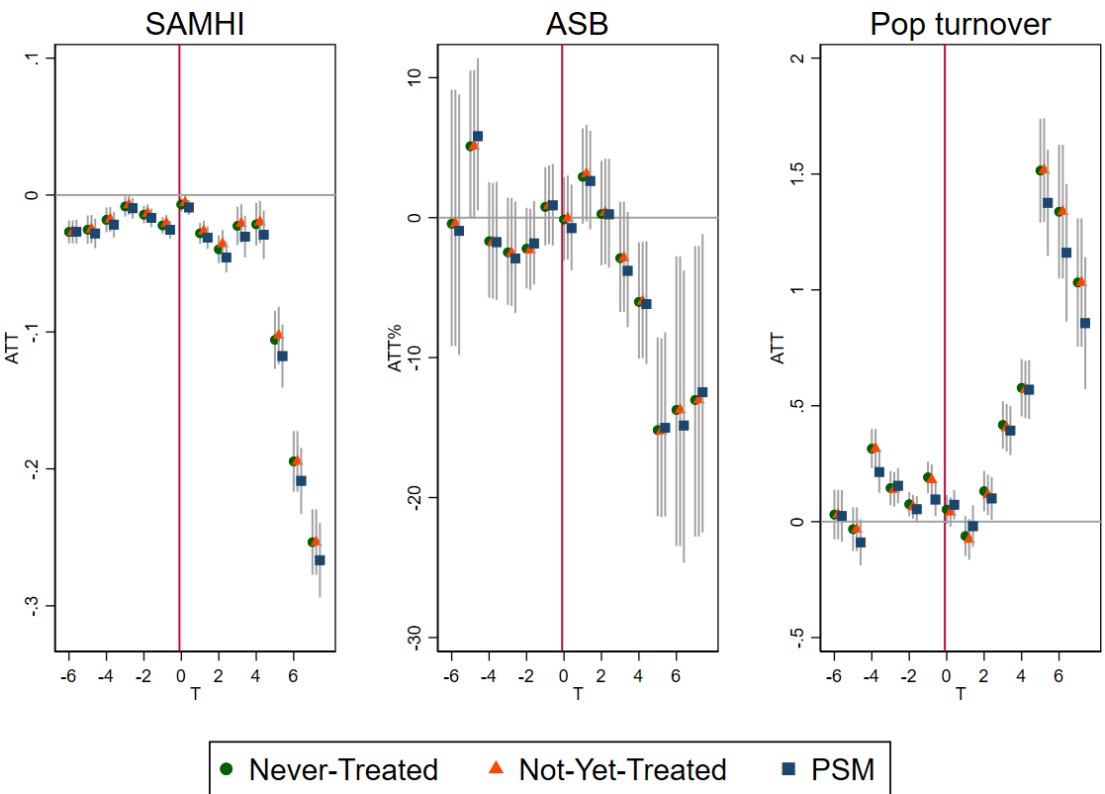

**Figure 3** Average treatment effect on the treated (ATT) for area-level impacts of selective licensing (SL) on Small Area Mental Health Index (SAMHI), Antisocial behaviour (ASB) calls and population (Pop) turnover in Greater London, 2011–2019. ASB was in-transformed and ATT shown as ATT%. PSM, propensity score matching.

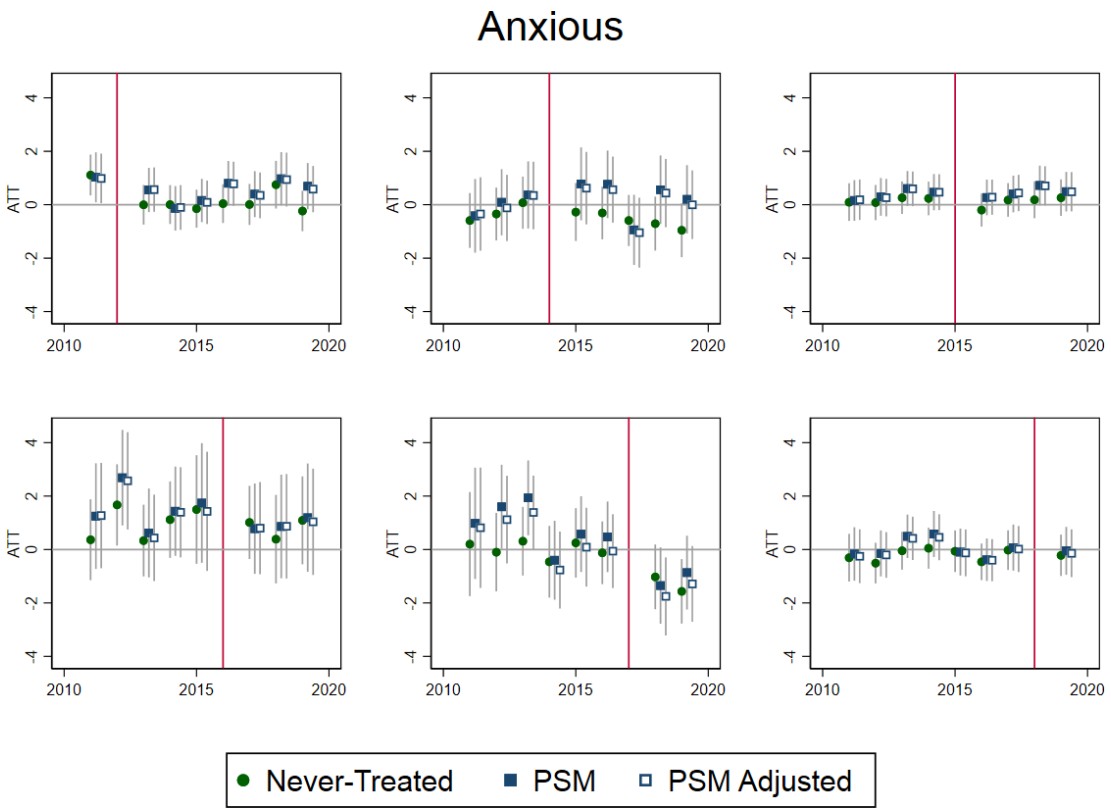

**Figure 4** Average treatment effect on the treated (ATT) for individual-level impacts of selective licensing (SL) on self-reported anxiety among private renters in Greater London by year of SL introduction, 2011–2019. Time-varying covariates in PSM adjusted were: age group, sex, native birth and occupational class. PSM, propensity score matching.

policies are a potential threat to the validity of our estimates.[36] Future research should, therefore, repeat our analysis when longer time series are available and more schemes can be studied in London and nationally to disentangle the effects of SL from the long-term effects of the urban regeneration such as those surrounding the London Olympics.

In this study, we defined mental health broadly with indicators ranging from self-reported well-being to mental health hospital admission. It is clear that the social surveys that cover subjective well-being are typically not designed for subregional analysis. Administrative or routinely collected data are, on the other hand, more scalable, yet only capture the more extreme end of the mental health scale, and often very hard to access for researchers due to information governance strictures. Recent developments triggered by the COVID-19 pandemi,c however, have opened up new opportunities for secure data linkage at patient address level.[37] This development is promising for the evaluation of housing policies such as SL.

A 10-year natural experiment study of healthcare service use in social housing residents age 60+ years in the UK found that those who received improvements to their kitchens, bathrooms or front doors, among other kinds of improvement, presented less often with common mental health disorders than those who did not receive these improvements.[8] A 5-year study (GoWell) of the impact of housing improvements on self-reported mental health and well-being among social housing residents found additional positive effects of renewing fabric works, that is, carpets, curtains and blinds.[6] The GoWell study also found a positive correlation between self-reported mental health and well-being among social housing residents and urban regeneration spending, which locally could cover internal housing, external housing, neighbourhoods and community project investments. It was the residents with the highest needs, who resided in the worst housing in the most rundown neighbourhoods, receiving the highest urban regeneration investment,who ultimately showed the greatest improvements in self-reported mental health.[7] Another UK natural experiment study of urban regeneration found positive effects for residents' mental health.[38]These studies support the link between housing improvement and mental health and well-being suggested by this study.

A recent systematic review on housing and health reported randomised controlled trial evidence about mental health benefits for both children and adults in relation to improvements of heating and ventilation.[2] Another recent systematic review of earlier housing disadvantage and poor mental health outcomes reported clear correlations, but also called for more studies to elucidate mechanisms.[39] Another review identified PRS as a growing yet overlooked sector with wide-ranging needs including mental health needs.[16] The review also acknowledged a current lack of evidence about effective interventions. Taken together, the reviews highlight a need for more

and better evidence of social polices aiming to improve housing quality including in PRS.

Reduction of ASB[18] is considered a key objective for the policing of London based on consultation and social surveys on the perception of crime.[40] It is common for local authorities to use reduction of ASB as a justification for SL[14] Interestingly, we found that ASB reduced after 4–5 years of SL—even when we excluded the 'Olympic' scheme. Further studies should examine the reasons for the ASB calls, for example, whether the calls concern neighbours.

A strength of the study is our use of the DiD design, which assesses impacts over and above a host of other factors that influence mental health and well-being. In addition, the multiple time period comparison DiD summarises the effect of a staggered intervention such as SL in a single analysis.[30] This step also enables not-yet treated as control of unmeasured factors associated with treatment allocation. Never-treated controls were included, should true effects be masked by overmatching in the PSM. Reassuringly, the different controls generally yielded similar results in this study.

The area-level findings should be backed up by individual-level findings specific to private renters and free of ecological bias.[41] In this case, we found that the APS sample data were too sparsely populated to create robust panel units over time and that many of the smaller schemes, therefore, could not be properly assessed. We instead deployed a canonical DiD approach and analysed SL by year of treatment initiation. The results were, however, inconclusive due to the large variation associated with small sample size. Future studies should include data at the national level to reach higher numbers.

A limitation of the study is that while physical housing conditions is a key factor in the logic model linking SL to more distant outcomes such as mental health and well-being, no adequate data were available to the authors at this point. We did consider national surveys such as English Housing Survey but assessed them too small for robust analysis, given the relatively sparse coverage of SL to date. We aim to address the important role of physical housing conditions in future studies, for example, by exploiting data from Energy Performance of Buildings Register or by linking administrative data on housing tenure to administrative healthcare data. We essentially call for more high-quality, data with sufficient temporal and spatial granularity to enable the timely evaluation of housing policies and their impact on both properties, people and localities.

We also call for a register of private rented properties and landlords to facilitate improved monitoring, evaluation and regulation of this sector. A recent UK government policy paper, a fairer PRS, has proposed a 'Property Portal', with landlords legally required to register their property on the portal.[42]

This study is to our knowledge the first to use SAMHI[23] and CDRC Residential Mobility index[25] in an evaluation of an area-based policy such as SL. There was much higher

precision in the SAMHI subscores, PRESCRIPTION and DIAGNOSIS, than in BENEFITS and ADMISSION. The results with ADMISSION were particularly unprecise and variable. CDRC Residential Mobility index provides yearly estimates of moves, whereas the 'gold standard', the Census flow data, in contrast are only released every 10 years.[43] The trend in annual proportion of households that will move in the coming year showed a great deal of fluctuation in itself. Due to the DiD design of this study, 'global' fluctuations are in themselves not prohibitive for an evaluation of an area-based intervention. Future releases should nonetheless examine whether the fluctuations can be explained.

The PSM used as far as possible preintervention sociodemographic and housing variables. It is possible that the matching could produce a more realistic counterfactual if more preintervention data relevant to treatment allocation and/or outcome risk factors become available in the future.

## CONCLUSIONS

We found early indications of a reduction in area-based mental health outcomes and ASB, while population turnover increased. Results from the individual-level analysis of APS data were inconclusive; possibly due to sample size issues. Longer time series are needed to disentangle SL from Olympic regeneration. Further studies specific to private renters and gentrification effects are needed. Overall, we argue that a national evaluation of SL is feasible and necessary.

**Author affiliations**
[1]Public Health, Environments & Society, London School of Hygiene and Tropical Medicine, London, UK
[2]Department of Public Health and Policy, University of Liverpool Faculty of Health and Life Sciences, Liverpool, UK
[3]Patient and Public Involvement Representative, London, UK
[4]School of Science & Technology, Department of Natural Sciences, Middlesex University, London, UK
[5]Head of Private Sector Housing, London Borough of Hackney, London, UK

**Contributors** All authors contributed to the conception, study design, data interpretation and approved the submitted version (JP, AA, DB, LC, EC, SC, DM, MS, JS, KT and ME). JP contributed to data acquisition, data analysis, drafted the first manuscript, and is the guarantor for the work.

**Funding** This study is funded by the National Institute for Health Research (NIHR) School for Public Health Research (SPHR) (Grant Reference Number PD-SPH-2015).

**Disclaimer** The views expressed are those of the authors and not necessarily those of the NIHR or the Department of Health and Social Care.

**Competing interests** None declared.

**Patient and public involvement** Patients and/or the public were involved in the design, or conduct, or reporting, or dissemination plans of this research. Refer to the Methods section for further details.

**Patient consent for publication** Not applicable.

**Ethics approval** Ethical approval was obtained from London School of Hygiene and Tropical Medicine's Ethics Committee (reference number 26481) and London Borough of Hackney.

**Provenance and peer review** Not commissioned; externally peer reviewed.

**Data availability statement** Data may be obtained from a third party and are not publicly available. The data supporting the findings of this study were obtained under licence and as such not available to other researchers. The data are, however, available from Office for National Statistics subject to ethical and scientific approval. This work was produced using statistical data from ONS (Annual Population Survey). The use of the ONS statistical data in this work does not imply the endorsement of the ONS in relation to the interpretation or analysis of the statistical data. This work uses research datasets which may not exactly reproduce National Statistics aggregates.

**ORCID iDs**
Jakob Petersen http://orcid.org/0000-0002-6659-7028
Laura Cornelsen http://orcid.org/0000-0003-3769-8740
Steven Cummins http://orcid.org/0000-0002-3957-4357
Maureen Seguin http://orcid.org/0000-0003-4786-9400

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
