## [Reviewer comments · BMJ Open]

ARTICLE DETAILS

TITLE (PROVISIONAL)	Impact of selective licencing schemes for private rental housing on mental health and social outcomes in Greater London, England: a natural experiment study
AUTHORS	PETERSEN, JAKOB; Alexiou, Alexandros; Brewerton, David; Cornelsen, Laura; Courtin, Emilie; Cummins, Steven; Marks, Dalya; Seguin, Maureen; Stewart, Jill; Thompson, Kevin; Egan, Matt

VERSION 1 – REVIEW

REVIEWER	Harris, Jennifer University of Bristol
REVIEW RETURNED	20-Sep-2022

GENERAL COMMENTS	- It's good to see such a rare attempt to measure the impact of regulatory interventions in the private rented sector. Would be great to see the study replicated on a larger scale to determine if the effects observed are specific for private renters. - In the introduction the authors focus on ASB as a justification for introducing SL schemes by local authorities. There were initially two conditions by which an authority was able to make a designation: area of low housing demand or the area is experiencing a significant and persistent problem caused by anti-social behaviour. Further conditions were added in 2015, to include housing conditions, migration, deprivation, and crime. The independent review of SL identified poor property conditions as the main reason cited by local authorities for introducing the schemes closely followed by ASB. Given the article's focus on health I think objectives related to property conditions require more recognition by the authors - Did the authors consider exploring if other objectives were achieved by the schemes? Particular those related to improved housing conditions or the reduction in hazards which is likely to improve mental health? If the authors ruled out a focus on other outcomes because of a lack of geographically specific/administrative housing data, then this would be helpful to know. In the private rented sector, it is very difficult to understand what is going on at a geographically specific area without more granular data. Do the authors feel that more needs to be done on providing coherent and robust statistics on the private rented sector so that we can look at other outcomes that the schemes aim to achieve – specifically area-wide improvements in housing conditions across geographies and at the local level? At the end the authors very briefly mention a need to explore links between administrative housing data and administrative health care data.
--

	An example of the administrative housing data the authors are referring to would be helpful. Is administrative housing data in its current form sufficient to allow us to do this? If not what needs to change (e.g. more granular data that can be aggregated?) - The authors mention in a couple of places that the mechanisms by which SL leads to improvements in ASB is not clear. However, I believe we do have some anecdotal knowledge regarding the mechanisms by which these effects occur – for example, training, information or support schemes provided by local authorities under SL schemes may help landlords to intervene more effectively. A rationale for using ASB as justification for introducing SL is that landlords are failing to intervene where it is occurring and often it's a condition under each license that landlords must take reasonable action to prevent and reduce ASB. SL may also facilitate joint working across different services operating across a particular area to help tackle the underlying issues associated with ASB. The authors may wish to reconsider this statement and/or associated hypothesis. - I would like to see some recognition of the differences in ways in which schemes are designed and implemented across different local authorities as this will undoubtedly have a impact on their effectiveness. There are concerns that some schemes as currently implemented are not reaching their full potential. Is this something that needs to be reflected on? Does it have any implications for a study that aggregates results across a area or indeed nationally?
--	---

REVIEWER	Bentley, Rebecca University of Melbourne
REVIEW RETURNED	27-Sep-2022

GENERAL COMMENTS	This paper is excellent. It addresses a very important public policy question for health - does regulation of private rentals have the potential to improve population health? Authors have provided a detailed description of the analyses and their analytical design is thoughtful and comprehensive. The authors have identified and described the papers limitations. I have the following minor comments:  1. In the discussion, authors suggest that population turnover may indicate a process of gentrification. This may explain the findings in relation to ASB. The paper is not clear on whether authors included population turnover as a proxy for gentrification at the outset or if this emerged as a possible explanation for the findings once the study had been conducted. Please review the sections relating to population turnover and more explicitly articulate the purpose of including this outcome within the conceptual framework of the paper. 2. Please provide more detail on the intervention. Have SL schemes been implemented in similar ways across local areas? Have they been evaluated in terms of improvements in housing conditions? Do they change over time? Is it well known to the renting population which areas have SL (i.e., might people move to or from an area based on the introduction of the scheme)? Does the cost of renting vary/change?
--

VERSION 1 – AUTHOR RESPONSE

Reviewer 1

1. It's good to see such a rare attempt to measure the impact of regulatory interventions in the private rented sector. Would be great to see the study replicated on a larger scale to determine if the effects observed are specific for private renters.

Response

We thank the reviewer for this point. This is very encouraging for us as we are currently in the process of applying for funding for a national study.

2. In the introduction the authors focus on ASB as a justification for introducing SL schemes by local authorities. There were initially two conditions by which an authority was able to make a designation: area of low housing demand or the area is experiencing a significant and persistent problem caused by anti-social behaviour. Further conditions were added in 2015, to include housing conditions, migration, deprivation, and crime. The independent review of SL identified poor property conditions as the main reason cited by local authorities for introducing the schemes closely followed by ASB. Given the article's focus on health I think objectives related to property conditions require more recognition by the authors

Response

These are important points on a) legislative history, and b) physical property conditions.

- a) We now provide details in the introduction about the initial conditions for SL in the Housing Act 2004 (enacted from 2006) and the additional conditions that opened up with the 2015 legislation (for reference, the legislative history can also be found in the published protocol paper).
- b) We did consider data sources on changes in the physical conditions of rental housing over time. Selective licencing has to date only affected a relatively small proportion of the housing stock and we assessed national surveys such as English Housing Survey too small for a robust analysis. In the revised manuscript, we highlight in the Limitations that no suitable data source was available to us at this point. We are however considering using administrative data, e.g. from the Energy Performance of Buildings Register. This work would fit better in a new funding round as it would take time to familiarise ourselves with it, and properly assess the strengths and limitations for the application at a national level. For instance, one of the caveats that we would have to address is the dynamic nature of the register and any implications for evaluations of this nature. We also state our ambition to develop this part of the work together with linkage between administrative housing tenure data and administrative healthcare data.

3. Did the authors consider exploring if other objectives were achieved by the schemes? Particular those related to improved housing conditions or the reduction in hazards which is likely to improve mental health? If the authors ruled out a focus on other outcomes because of a lack of geographically specific/administrative housing data, then this would be helpful to know. In the private rented sector, it is very difficult to understand what is going on at a geographically specific area without more granular data. Do the authors feel that more needs to be done on providing coherent and robust statistics on the private rented sector so that we can look at other outcomes that the schemes aim to achieve – specifically area-wide improvements in housing conditions across geographies and at the local level? At the end the

authors very briefly mention a need to explore links between administrative housing data and administrative health care data. An example of the administrative housing data the authors are referring to would be helpful. Is administrative housing data in its current form sufficient to allow us to do this? If not what needs to change (e.g. more granular data that can be aggregated?)

Response

Unfortunately, while we did consider other objectives, we weren't able to explore these in this study. We agree with the reviewer that evaluations of housing policies in England are currently limited by the lack of sufficiently granular data. We see the reviewer's point as encouragement to developing better and more granular data resources as well as reaching more detailed analyses. As outlined in the response above, we consider administrative data linkage and disaggregation necessary. Examples of administrative housing data include Energy Performance of Buildings Register, the HM Land Registry, housing tenure classification data (owned by BRE Group), and the proposed national private landlord register. Locally, local authorities with SL schemes are collecting their own data on tenure and housing conditions through inspections, which we could request access to. There is likely to be considerable variation in the inspection data across the country and central data resources would therefore still be necessary for proper evaluation.

We have elaborated on this point in the revised Discussion and call for more high-quality data with sufficient temporal and spatial granularity to enable the timely evaluation of housing policies and their impacts on both properties, people, and localities. We have also referenced the UK government's recent proposal for a 'Property Portal'; with landlords legally required to register their property on the portal.

4. The authors mention in a couple of places that the mechanisms by which SL leads to improvements in ASB is not clear. However, I believe we do have some anecdotal knowledge regarding the mechanisms by which these effects occur – for example, training, information or support schemes provided by local authorities under SL schemes may help landlords to intervene more effectively. A rationale for using ASB as justification for introducing SL is that landlords are failing to intervene where it is occurring and often it's a condition under each license that landlords must take reasonable action to prevent and reduce ASB. SL may also facilitate joint working across different services operating across a particular area to help tackle the underlying issues associated with ASB. The authors may wish to reconsider this statement and/or associated hypothesis.

Response

This is a good point – and we agree that the mechanisms the reviewer suggests are plausible and have received some coverage in the literature. The published Independent Review on SL includes stakeholder opinions and examples that align with the potential mechanisms that the reviewer has suggested (Lawrence & Wilson 2019). Our own forthcoming qualitative paper will also consider mechanisms based on data we collected from Greater London. Therefore, we have revised the Introduction to include these points in more detail.

5. I would like to see some recognition of the differences in ways in which schemes are designed and implemented across different local authorities as this will undoubtedly have a impact on their effectiveness. There are concerns that some schemes as currently implemented are not reaching their full potential. Is this something that needs to be reflected

on? Does it have any implications for a study that aggregates results across a area or indeed nationally?

Response

An interesting finding of our paper is that changes in outcomes appeared to occur before the completion of the 5-year licensing period. It raises the possibility that SL schemes do not need to be at their 'full potential' to have an area level impact. This is important because (as the reviewer correctly indicates) some critics of SL have argued that SLs are unlikely to be effective due to incomplete implementation (e.g. it may not be possible to ensure all landlords join the scheme, all homes are inspected, and all mandated improvements are carried out). We would suggest that few interventions of any kind really achieve this kind of 'full' implementation, but they may still do enough to make an impact. We have now briefly added this point to the paper in the Discussion (p. 9).

With regards to the issue that different SL schemes are implemented in different ways, we originally alluded to this in a limited way, for example by pointing out that they varied in size. We agree with the reviewer that it is worth giving some additional detail – which we have now done in the Introduction (pp. 3-4) – outlining different approaches to tackling ASB along the lines suggested by the Reviewer in comment 4. Nonetheless, the study does not seek to evaluate each specific scheme so there are limits to how much we want to go into the workings of each of them. The Discussion does now refer to variation in the levels of enforcement across London schemes (p. 9). While there has not been a robust evaluation of this issue, the website www.londonpropertylicensing.co.uk does include some data on this based on periodic information requests from London local authorities. We now allude to this.

Reviewer 2

This paper is excellent. It addresses a very important public policy question for health - does regulation of private rentals have the potential to improve population health? Authors have provided a detailed description of the analyses and their analytical design is thoughtful and comprehensive. The authors have identified and described the papers limitations.

1. In the discussion, authors suggest that population turnover may indicate a process of gentrification. This may explain the findings in relation to ASB. The paper is not clear on whether authors included population turnover as a proxy for gentrification at the outset or if this emerged as a possible explanation for the findings once the study had been conducted. Please review the sections relating to population turnover and more explicitly articulate the purpose of including this outcome within the conceptual framework of the paper.

Response

We mention 'residential stability' as a proxy in relation to the individual level analyses and for consistency we now also explain that the residential mobility index is deployed as a proxy for moves in the area-based analyses. Several mechanisms could potentially be at play (now outlined in the Discussion). SL may result in more landlords selling their properties rather than facing the increased cost burden, increasing illegal rentals, passing costs onto tenants through rent increases, and evicting tenants with ASB behaviours; with the opportunity in increase rents in high-demand areas. Our current funding application for a national study proposes to look at whether rents go up. We will in this way be able to narrow in on any gentrification effects.

2. Please provide more detail on the intervention. Have SL schemes been implemented in similar ways across local areas? Have they been evaluated in terms of improvements in housing conditions? Do they change over time? Is it well known to the renting population which areas have SL (i.e., might people move to or from an area based on the introduction of the scheme)? Does the cost of renting vary/change?

Response

We have now added more information making clear that there can be variation in implementation, as well as contextual variation (see response to reviewer 1 – and also revisions on pp. 3,4 and 9.)

As with reviewer 1, reviewer 2 correctly identifies that data on (change in) housing conditions would be extremely useful to this study. We now discuss more fully this issue, and include a call for improved data. The proposed national study does include analysis of rental costs over time but it is outside the parameters of the current study. Also, see our response to Reviewer 1, comment 2, where we discuss some of the issues with data on housing conditions, and we explain that we plan to include potentially useful data, such as those from the Energy Performance of Buildings Register, in the proposed national study.

As a companion piece to the current manuscript, we plan to soon submit a subsequent paper based on qualitative data (stakeholder interviews). Our qualitative data suggests that landlords do not always know about the scheme. Furthermore, some of the research team accompanied housing inspectors in one scheme and it was certainly clear that many tenants in the area did not know about the scheme. We will discuss this more in the subsequent qualitative publication. We have not heard of residents choosing to move specifically so that they can be in a SL area (our guess would be that it is doubtful or, at best, rare). We do hypothesise in this revised paper that the scheme could oblige tenants to move away if they are evicted or if rents rise because of the way landlords react to the scheme.

In short, there are numerous issues related to the mechanisms that we do not yet know, and some that will be reported in a subsequent article. Our next (qualitative paper) will provide evidence on some mechanisms, but we believe a larger national study is required to understand more.